# Monte Carlo Simulations for the Estimation of the Effective Permeability of Mixed-Matrix Membranes

**DOI:** 10.3390/membranes12111053

**Published:** 2022-10-27

**Authors:** Zheng Cao, Boguslaw Kruczek, Jules Thibault

**Affiliations:** Department of Chemical and Biological Engineering, University of Ottawa, Ottawa, ON K1N 6N5, Canada

**Keywords:** mixed-matrix membrane (MMM), relative permeability, solubility, Monte Carlo simulations, finite difference simulations

## Abstract

Recent years have seen the explosive development of mixed-matrix membranes (MMMs) for a myriad of applications. In gas separation, it is desired to concurrently enhance the permeability, selectivity and physicochemical properties of the membrane. To help achieving these objectives, experimental characterization and predictive models can be used synergistically. In this investigation, a Monte Carlo (MC) algorithm is proposed to rapidly and accurately estimate the relative permeability of ideal MMMs over a wide range of conditions. The difference in diffusivity coefficients between the polymer matrix and the filler particle is used to adjust the random progression of the migrating species inside each phase. The solubility coefficients of both phases at the polymer–filler interface are used to control the migration of molecules from one phase to the other in a way to achieve progressively phase equilibrium at the interface. Results for various MMMs were compared with the results obtained with the finite difference method under identical conditions, where the results from the finite difference method are used in this investigation as the benchmark method to test the accuracy of the Monte Carlo algorithm. Results were found to be very accurate (in general, <1% error) over a wide range of polymer and filler characteristics. The MC algorithm is simple and swift to implement and provides an accurate estimation of the relative permeability of ideal MMMs. The MC method can easily be extended to investigate more readily non-ideal MMMs with particle agglomeration, interfacial void, polymer-chain rigidification and/or pore blockage, and MMMs with any filler geometry.

## 1. Introduction

In the last 40 years, membrane technology for gas separation has seen tremendous growth in the number of industrial applications where, in many cases, it can now compete advantageously with conventional unit operations such as distillation, absorption and adsorption. More recently, mixed-matrix membranes (MMMs), where a filler is incorporated within the matrix of a polymeric membrane, have seen tremendous research interest and an increase in the number of potential applications [1]. MMMs have been proposed in order to modify the characteristics of the membrane to improve the desired performance, making use of the advantages stemming from the synergistic characteristics of both polymeric and inorganic membranes. The last decade has seen significant developments in MMMs with numerous applications: gas separation with functionalized fillers [2], ethanol dehydration [3], ABE solvent separation [4,5], treatment of oil sands produced water [6], water desalination [7], embedding environmentally friendly nanoclay palygorskite for flux enhancement and fouling reduction [8] and many others. The main benefits of MMMs are the process intensification and the performance enhancement by the presence of suitable fillers while preserving the economic and processing convenience of the polymer [9].

The most common use of MMMs in gas separation and solvent pervaporation has targeted the change in the permeability and selectivity of the membranes. For instance, to improve the permeability of the membrane, it is possible to incorporate higher performance nanoporous or high-permeability fillers where preferential diffusion paths are created within the MMM. On the other hand, if the purpose of the membrane is to act as a barrier material, impermeable fillers can be embedded in the polymer matrix to increase the tortuosity and reduce the membrane’s effective permeability. The permeability of the polymer, the filler and the MMM is the product of their respective diffusivity and solubility. It has been clearly shown through correlations and numerical analysis that the relative effective permeability of an MMM is impacted by the relative permeability of the two phases (filler and polymer) and not by their individual relative diffusivity and relative solubility [10]. This is contrary to the results presented by Singh et al. [11], who argued that both the diffusivity ratio and the solubility ratio should be considered separately. Most empirical models and numerical models used for the estimation of the relative permeability of MMMs considered ideal MMMs assuming the absence of interfacial voids and rigidified polymeric regions at the interface with the filler, in addition to assuming that the filler is uniformly distributed in the polymer matrix and that the polymer does not penetrate the pores of the fillers. It is obvious that, in practice, most MMMs are not ideal, and the effective permeability may be different. Nevertheless, it is important to estimate the effective permeability of ideal MMMs such that any deviations from ideality observed experimentally can be better assessed and understood. Many analytical models are available in the literature and have been extensively used by many researchers. However, these models are limited to fillers of simple geometrical shapes, most often spherical. Numerical techniques such as the finite differences method (FDM) [4,12] and finite element method (FEM) [11,13] have been extensively used to calculate the relative permeability of MMMs and considered to be the actual values.

To manufacture and optimize an MMM with the desired properties for a specific gas separation, it is paramount to gain a deeper understanding of the transport properties of these MMMs. The synergy of modelling and experimentation can play an essential role in attaining these objectives. In the past decade, our group has used numerical simulations to determine the effective permeability of ideal MMMs embedding filler particles having different diffusivity and/or solubility than the host polymer, thereby decreasing or enhancing the effective permeability of the MMM. These numerical simulations were performed by solving the three-dimensional second Fick’s law of diffusion using the FDM. The FDM requires complex coding and significant computation time. In search of a more straightforward simulation method to implement, we have opted for a Monte Carlo (MC) algorithm. The principle of an MC simulation is to approximate the expected value of a random variable by an average over a large number of measurements. There are different scales at which simulations of the diffusion process can be performed, going from the atomistic scale [14,15,16] by considering a very tiny evaluation space-domain to the macroscopic scale by solving the Fick’s second law of diffusion for the whole membrane using a suitable solver for partial differential equations [4]. Atomistic simulations consider stretching, bending and torsional motion of the polymer chain and permeating molecules, which allows studying adsorption and diffusion locally. This degree of granularity is not useful for estimating the relative permeability of MMMs. As an intermediate between these two opposing scales, this investigation proposes to use an MC approach to indirectly solve Fick’s second law of diffusion by simply considering the Brownian motion of molecules within an MMM. The MC algorithm is much easier to implement and should provide the same level of accuracy akin to more conventional solutions of the diffusion process.

## 2. Methodology

### 2.1. Definition of the Simulation Domain

In this investigation, it is assumed that the MMM is ideal, implying that all solid filler particles, having identical shape, geometry and orientation, are uniformly distributed within the polymer matrix. In addition, the polymer/particle interface is considered ideal, and the permeability and solubility coefficients for both the polymer and the filler particles are assumed constant. The two-phase MMM is subdivided into a number of elementary units. Each elementary unit consists of a polymer matrix with a centrally located solid particle schematically presented in Figure 1. The solid volume fraction of an elementary unit is identical to the volume fraction of the entire MMM. It has been shown that the effective permeability of an elementary unit is identical to the permeability of the entire MMM [10,17]. As a result, to solve numerically for the effective permeability of MMMs, it is only necessary to consider one elementary unit. For all finite difference (FD) and Monte Carlo (MC) simulations, in Cartesian coordinates, the three-dimensional elementary unit of dimensions *L_x_* × *L_y_* × *L_z_* is discretized into a grid *n_x_* × *n_y_* × *n_z_* finite volumes. The solid filler particle of dimensions *x_F_* × *y_F_* × *z_F_* located at the center of the elementary unit has a diffusivity coefficient (*D_F_*) and a solubility coefficient (*S_F_*), whereas, for the polymer in the elementary unit surrounding the filler, they are *D_p_* and *S_p_*, respectively.

The initial and boundary conditions need to be defined to solve this problem. For providing the boundary conditions, the filler particle is assumed to be a cuboid, as defined in the previous paragraph. For the initial condition, that is, before the concentration step change on one side of the membrane occurs at time *t* = 0, a nil concentration is assumed throughout the membrane (Equation (1)). For the boundary conditions in Cartesian coordinates, twelve relations are required to completely define the problem: six boundary conditions at the periphery of the polymeric elementary unit and six boundary conditions at the polymer–solid interfaces. Equation (2) provides the boundary conditions on the upstream and downstream sides of the membrane (the *y*-axis is the permeation direction). It is assumed that at the onset of permeation, a step change in the gas pressure or concentration is applied to the upstream side of the membrane (*y* = 0), whereas the gas pressure on the downstream side of the membrane (*y* = *L_y_*) is kept under perfect vacuum. The resulting concentrations in the membrane at both surfaces are simply the product of the neighboring gas pressure or concentration and the solubility *S_P_* of the polymer. As all elementary units are identical, symmetry conditions prevail at the other four faces of the polymer parallelepiped as expressed in Equation (3) for BC_3–6_. These are periodic boundary conditions since the MMM is formed of a large number of identical elementary units. At the filler particle–polymer interface, it is assumed that equilibrium between the two phases prevails at each of the six faces of the nanoparticle (see BC_7–12_ in Equation (4)). Even though Equation (4) applies for cuboid particles aligned with the membrane surface, similar boundary conditions can easily be defined for other particle geometry. More details on the derivation and solution of these equations can be found in Wu et al. [18,19]. The results obtained by the FDM are considered true values and will be used as the benchmark for evaluating the accuracy of the results obtained by the Monte Carlo simulations.
(1)$$\begin{array}{cc} \mathrm{IC}: & C_{i,j,k}^{t<0}=0 \end{array}\;\forall i,j,k$$
(2)$$\begin{array}{l} {\mathrm{BC}}_{1}:\;C_{i,y\;=\;0,k}=\frac{p_{0}}{RT}S{=c}_{0}S\;\;\forall i,k\\ {\mathrm{BC}}_{2}:\;C_{i,y\;=\;L_{y},k}=0\;\;\;\;\;\forall i,k \end{array}$$
(3)$${\mathrm{BC}}_{3–6}:\;{\left.\frac{\partial C}{\partial x}\right|}_{x\;=\;0}={\left.\frac{\partial C}{\partial x}\right|}_{x\;=\;L_{x}}={\left.\frac{\partial C}{\partial z}\right|}_{z\;=\;0}={\left.\frac{\partial C}{\partial z}\right|}_{z\;=\;L_{z}}=0$$



(4)
$$\begin{array}{c} {\mathrm{BC}}_{7}:C_{\left(L_{x}-x_{p}\right)/2,y,z}^{\mathrm{Filler}\;}=\frac{S_{p}}{S_{F}}C_{\left(L_{x}-x_{p}\right)/2,y,z}^{\mathrm{Polymer}\;}\;\forall \left[\left(\frac{\left(L_{y}-y_{p}\right)}{2}\leq y\leq \frac{\left(L_{y}+y_{p}\right)}{2}\right)\cap \left(\frac{\left(L_{z}-z_{p}\right)}{2}\leq z\leq \frac{\left(L_{z}+z_{p}\right)}{2}\right)\right]\\ {\mathrm{BC}}_{8}:C_{\left(L_{x}+x_{p}\right)/2,y,z}^{\mathrm{Filler}\;}=\frac{S_{p}}{S_{F}}C_{\left(L_{x}+x_{p}\right)/2,y,z}^{\mathrm{Polymer}\;}\;\forall \left[\left(\frac{\left(L_{y}-y_{p}\right)}{2}\leq y\leq \frac{\left(L_{y}+y_{p}\right)}{2}\right)\cap \left(\frac{\left(L_{z}-z_{p}\right)}{2}\leq z\leq \frac{\left(L_{z}+z_{p}\right)}{2}\right)\right]\\ {\mathrm{BC}}_{9}:C_{x,\left(L_{y}-y_{p}\right)/2,z}^{\mathrm{Filler}\;}=\frac{S_{p}}{S_{F}}C_{x,\left(L_{y}-y_{p}\right)/2,z}^{\mathrm{Polymer}\;}\;\forall \left[\left(\frac{\left(L_{x}-x_{p}\right)}{2}\leq x\leq \frac{\left(L_{x}+x_{p}\right)}{2}\right)\cap \left(\frac{\left(L_{z}-z_{p}\right)}{2}\leq z\leq \frac{\left(L_{z}+z_{p}\right)}{2}\right)\right]\\ {\mathrm{BC}}_{10}:C_{x,\left(L_{y}+y_{p}\right)/2,z}^{\mathrm{Filler}\;}=\frac{S_{p}}{S_{F}}C_{x,\left(L_{y}+y_{p}\right)/2,z}^{\mathrm{Polymer}\;}\;\forall \left[\left(\frac{\left(L_{x}-x_{p}\right)}{2}\leq x\leq \frac{\left(L_{x}+x_{p}\right)}{2}\right)\cap \left(\frac{\left(L_{z}-z_{p}\right)}{2}\leq z\leq \frac{\left(L_{z}+z_{p}\right)}{2}\right)\right]\\ {\mathrm{BC}}_{11}:C_{x,y,\left(L_{z}-z_{p}\right)/2}^{\mathrm{Filler}\;}=\frac{S_{p}}{S_{F}}C_{x,y,\left(L_{z}-z_{p}\right)/2}^{\mathrm{Polymer}\;}\;\forall \left[\left(\frac{\left(L_{x}-x_{p}\right)}{2}\leq x\leq \frac{\left(L_{x}+x_{p}\right)}{2}\right)\cap \left(\frac{\left(L_{y}-y_{p}\right)}{2}\leq y\leq \frac{\left(L_{y}+y_{p}\right)}{2}\right)\right]\\ {\mathrm{BC}}_{12}:C_{x,y,\left(L_{z}+z_{p}\right)/2}^{\mathrm{Filler}\;}=\frac{S_{p}}{S_{F}}C_{x,y,\left(L_{z}+z_{p}\right)/2}^{\mathrm{Polymer}\;}\;\forall \left[\left(\frac{\left(L_{x}-x_{p}\right)}{2}\leq x\leq \frac{\left(L_{x}+x_{p}\right)}{2}\right)\cap \left(\frac{\left(L_{y}-y_{p}\right)}{2}\leq y\leq \frac{\left(L_{y}+y_{p}\right)}{2}\right)\right] \end{array}$$



### 2.2. Monte Carlo Algorithm

The MC method was developed during the Manhattan Project for modelling neutron diffusion during the fission process and is now used in a myriad of applications in numerous fields of study [20,21,22]. In this investigation, the dynamic MC simulations are used to model the dynamic behavior of molecules permeating through an MMM where all molecules are subjected to Brownian diffusion or random walk diffusion. In this investigation, the implementation of the MC was performed using two different methods. In the first method, referred to as MC1, the displacement of molecules occurs in a cubic lattice, where a molecule can only jump from the current mesh element (*i*, *j*, *k*) to one of the six neighboring meshes (Figure 2). In the second method, referred to as MC2, the displacement of molecules can occur in any direction with a jump proportional to the local relative diffusivity. The relative diffusivity is calculated as the diffusivity of the phase where the molecule under consideration is currently located, divided by the highest diffusivity of the system, i.e., the diffusivity of the polymer or the filler particle. In MC1, the molecules are located only within one of the mesh volume elements. In contrast, for MC2, molecules can assume any location within the elementary unit and their (*x*, *y*, *z*) coordinates are recorded throughout their migration through the membrane. To avoid redundancy, the description of the MC method will be focused on the cubic lattice method (MC1), and only minor differences for the second method will be provided. It is important to mention that it is not possible for the macroscopic aspect of the MC, as used in this investigation, to determine the effective membrane permeability directly. Instead, the relative membrane permeability (*P_r_* = *P_e_*/*P_P_*), that is, the ratio of the effective permeability of the MMM over the permeability of the neat polymeric membrane, is determined by performing the identical MC simulation and comparing the steady-state permeation flux of an MMM with the steady-state flux of an identical membrane without the presence of a filler particle. Since the diffusivity, solubility and permeability of the neat membrane are usually known or can be more easily determined experimentally, the effective permeability of the MMM can, therefore, be estimated. In this investigation, the results of an MC simulation for an MMM will always be compared with the results of a neat membrane under identical conditions.

Simulations were performed with an MMM having a nil initial concentration of molecules throughout the membrane. At the onset of the dynamic permeation experiment, a step change in the gas pressure or gas concentration on the upstream side of the membrane is applied, which leads to a sudden increase in the number of molecules diffusing into the upstream surface layer of the membrane. It is assumed that there is an instantaneous equilibrium at the upstream interface. To maintain the concentration of molecules constant at the upstream interface of the membrane, it will be reset to the same value after each iteration, which will naturally set the flux of molecules entering the membrane. In gas permeation in a dense membrane, two main steps are considered: (1) the sorption of gases in the membrane material and (2) the diffusion of gas molecules from a region of higher chemical potential (concentration) to a region of lower chemical potential. MC simulation can create the permeation of molecules through the membrane due to the Brownian movement of molecules. Naturally, molecules move in a random walk from the upstream region (higher concentration) to the downstream region of the membrane (lower concentration). In MC simulations, each molecule within the membrane is subjected to a displacement at each iteration to mimic the molecule Brownian diffusion.

After discretizing the elementary unit with a large number of meshes, providing the values of the membrane characteristics and process variables and populating the upstream surface layer with a sufficiently large number of molecules, each iteration follows the same procedure as follows: (Figure 3 shows the pseudocode of the MC algorithm):

(a)All members of the current population are, in turn, subjected to a random move in one of the six adjoining meshes (Figure 2) such that a molecule will potentially move to an adjacent mesh. To consider the difference in the diffusivity coefficient between the two phases of the MMM, the probability of a molecule adopting the suggested move is made equal to the ratio of the diffusivity at the current mesh location and the maximum diffusivity value as expressed in Equation (5).
(5)$$IF\;\mathrm{RN}\;<\;\frac{D_{Current\;Mesh}}{Max(D_{F},D_{P})}THEN\;Move\;accepted$$If the random number (RN) is smaller than the diffusivity ratio, the move is considered. A move is always suggested but not necessarily realized depending on some additional considerations when the molecule under consideration is currently located within the phase having the lowest diffusivity. For instance, assuming a diffusivity ratio of 0.1, if the molecule is located in the phase with the lowest diffusivity, the probability of moving to a neighboring mesh slot is 0.1. If a molecule is currently located in the phase having the highest diffusivity, the move is always accepted provided few additional conditions are satisfied.(b)Indeed, when a move of a molecule is accepted, this particular move is subjected to two additional conditions to be realized.(i)If a molecule is currently located at one of the four lateral faces perpendicular to the *x*- and *z*-direction of the elementary unit and the move brings that molecule outside the elementary unit, then this molecule simply re-enters the elementary unit directly on the opposite face in order to satisfy the four symmetrical periodic conditions in the *x*- and *z*-directions [23].(ii)Since it is assumed that equilibrium prevails at the polymer–solid interface, if the suggested random move is from phase 1 to phase 2 (Figure 4), the move will be accepted if it favors the attainment of the equilibrium between the two phases. If the two solubility coefficients differ, a discontinuity in concentration will prevail at the polymer–solid interface. If the current ratio of concentration *C*_2_/*C*_1_ already exceeds the solubility ratio *S*_2_/*S*_1_, the move is not performed (Equation (6)). If the ratio of concentration following the move remains smaller than the solubility ratio, the move is performed (Equation (7)). Finally, if the concentration ratio before the move is smaller than the solubility ratio whereas it becomes greater after the move, the probability of accepting the move is calculated as given in Equation (8). It is important to point out that the interface equilibrium cannot be specified directly, but must be attained progressively through molecule random walk. Equations (6)–(8) ensure that the system is always trying to achieve equilibrium.
(6)$$\mathrm{IF}\;\frac{C_{2}}{C_{1}}\;\geq \;\frac{S_{2}}{S_{1}}\;\mathrm{THEN}\;No\;Move$$
(7)$$\mathrm{ELSE}\;\mathrm{IF}\;\frac{C_{2}+1}{C_{1}-1}\;\leq \;\frac{S_{2}}{S_{1}}\;\mathrm{THEN}\;Move$$
(8)$$\mathrm{ELSE}\;\mathrm{IF}\;\left(\frac{C_{2}}{C_{1}}<\frac{S_{2}}{S_{1}}\;\mathrm{AND}\;\frac{C_{2}+1}{C_{1}-1}>\frac{S_{2}}{S_{1}}\right){\;\mathrm{THEN}\;\mathrm{Prob}}_{\mathrm{Move}}=\frac{\frac{S_{2}}{S_{1}}-\frac{C_{2}}{C_{1}}}{\frac{C_{2}+1}{C_{1}-1}-\frac{C_{2}}{C_{1}}}$$(c)When all the members of the current molecule population have been considered in one given iteration, the population is updated. First, the number of molecules in the upstream surface layer is reset to its original number to satisfy the equilibrium condition between the constant surrounding gas pressure and the polymer surface of the MMM. Resetting the original number of molecules automatically calculates the flux of molecules entering the membrane at each iteration. As the Brownian motion is purely a random process, it is possible to encounter a negative flux for some iterations. Secondly, all the molecules that exited the downstream side of the membrane allow the calculation of the instantaneous permeation flux of molecules leaving the membrane. At each iteration, the molecules added or removed from the upstream membrane surface layer and the molecules exiting the downstream side serve to update the current population of molecules.(d)The above procedure is followed for a given number of iterations or until molecules’ average inlet and outlet fluxes are the same, i.e., the steady-state permeation is attained. The final results are then printed. The relative permeability of the MMM is calculated by dividing the steady-state permeation flux obtained for the MMM by the permeation flux of the neat membrane under identical conditions, implying that the conditions of Equation (5) are considered in calculating the flux of the neat membrane even if a filler is absent.(e)The above procedure pertains to the lattice MC. In the case of the second method alluded to earlier (MC2), the only difference is in calculating the displacement of the molecules permeating through the membrane. In the lattice MC, the molecules can jump from the central cubic mesh to one of the six neighboring meshes. In the second method, the molecules can move in any direction by randomly choosing a displacement in the three directions and calculating the new coordinates of a given molecule. In this method, a move is always accepted, but the distance travelled by one molecule is adjusted via the diffusivity ratio of the two phases, similar to what was performed in the lattice MC. All the other considerations, such as the periodic symmetrical conditions and the polymer–solid equilibrium interface, remain the same. For the latter two considerations and the upstream surface population, the number of molecules in each cubic mesh of the MMM is calculated and used in the same way as in the lattice MC.

The computer codes to perform MC and FD simulations were written in the FORTRAN programming language. When memory permitted, simulations were performed using a DELL Latitude 5500 laptop computer with an Intel Core i7-8665U CPU at 1.90 GHz and 16 GB installed RAM. For larger problems, the CAC cluster Frontenac at Queen’s University (Kingston Canada) was used, which comprised a wide variety of computers with large memory ranging from 128 GB to 2 TB. The advantage of the cluster is the possibility of submitting many simulations with larger populations at the same time.

## 3. Results and Discussion

The MC simulation is pretty robust in determining the relative permeability of MMMs because the migration flux for a particular MMM is compared with the flux in the neat polymeric membrane under identical conditions. Indeed, this procedure provides the relative effective permeability of the MMM and not the actual effective permeability. It is, therefore, not necessary to have the exact time scale to obtain this information as it is assumed that the same time is elapsed for each iteration. To explore the ability of the MC simulation to provide the relative permeability of the MMM, a series of tests was performed and compared to the same results obtained with the FD method, which is assumed to provide the actual value of the MMM permeability and relative permeability.

Figure 5 presents the inlet and outlet fluxes obtained during a typical MC simulation using method MC1. The initial population is nil throughout the membrane. At time zero, the first upstream surface layer is populated with a given number of molecules that is reset to a constant value at each iteration. The output flux is nil until some molecules from the higher upstream concentration migrate through the membrane to emerge at the downstream side of the membrane and eventually leading to an average constant outlet flux. The initial input flux is very high at the beginning because of the large gradient of molecules in the meshes close to the upstream surface of the membrane. There are initially very few molecules in the downstream portion of the membrane to create a backflow of molecules to the upstream portion of the membrane via the natural Brownian motion to significantly reduce the net flow of molecules entering the upstream membrane surface. As the population of molecules progressively builds up with the number of iterations, the average input flux stabilizes to reach a constant value equal to the output flux under steady-state conditions. The solid line in Figure 5 represents the moving average of the instantaneous input and output fluxes, the instantaneous fluxes represented by the light color plots. Figure 5 shows that the instantaneous flux varies significantly due to the random nature of the Brownian motion. The inlet flux varies much more than the output flux because of the much larger number of molecules present at the upstream membrane interface. For some iterations, the instantaneous input flux is negative because of the random walk backflow of molecules to the first upstream membrane layer. Since equilibrium exists between the surrounding gas and the upstream surface layer, the probability of molecules leaving the first layer to satisfy the equilibrium condition is not nil. The flux variation on the membrane’s downstream side is not as severe because the number of molecules is significantly less than on the upstream side. The MC simulation shows more realistically the phenomenon occurring in an actual gas permeation process across a membrane. In contrast, the solution the Fick’s second law of diffusion by FD assumes a direct permeation path from the upstream to the downstream sides of the membrane using diffusion coefficients smaller than the ones associated with the random walk diffusion.

The two MC methods, MC1 and MC2 discussed in Section 2.2, are compared under identical conditions for the same MMM having a cubic nanoparticle of different sizes. The plot of the relative steady-state flux or relative permeability of the MMM for three different levels of discretization is presented in Figure 6 and compared with the results obtained with the FD method. The two MC methods gave nearly identical results for the three levels of discretization. In addition, these results show that the accuracy of the MC methods is excellent with an average error of 0.008 in the estimation of the relative permeability and a maximum error of 0.015 at the higher volume fraction. In addition, the method is not sensitive to the discretization level, at least for cubic filler particles. In most of the results presented in this work, a 41 × 41 × 41 discretization grid was used to ensure maximum accuracy under all conditions.

Figure 7 presents the results obtained for MMMs with an impermeable cuboid filler as a function of the filler volume fraction for both the MC and FD methods. In each of the four series of simulations, the thickness (*y_F_*) of the cuboid filler remained constant while the other two dimensions of the filler (*x_F_* = *z_F_*) were progressively increased to augment the filler volume fraction and, as a result, the filler projected area. The four different values of the filler thickness were 5, 15, 25 and 35, defined on a three-dimensional grid of 41 × 41 × 41. The other two dimensions were varied from 5 to 37 by increments of 4. The results in Figure 7 clearly show that the MC method can predict the relative permeability of the MMM very accurately for the first three thicknesses even when only two cubic mesh volumes (*x_F_* = *z_F_* = 37) are available around the filler for the molecules to permeate to the downstream side of the membrane. Indeed, the absolute value of the error (AE), defined in Equation (9), are 0.006, 0.006, 0.008 and 0.022, respectively, for thicknesses 5, 15, 25 and 35. For the thicker filler (*y_F_* = 35), the average AE for the estimation of the relative permeability of the MMM is about three times higher than for the other three thicknesses. This latter situation is obviously an extreme case since the thickness of the particle represents 85% of the thickness of the elementary unit. In general, and based on these results, it can be concluded that MC simulations can be used with confidence for the estimation of the relative permeability of MMMs.
(9)$$AE=\left|P_{r,MC}{\;-\;P}_{r,FD}\right|$$

The results of Figure 7 highlight a very important aspect concerning estimating the relative permeability of ideal MMMs. There exists a significant difference in the values of the relative permeability for an identical filler volume fraction. The most popular correlations for the estimation of the relative permeability of MMMs use the volume fraction of the filler as the main model parameter and, therefore, cannot account for the significant variations observed in the results in Figure 7. This is the case for the well-known correlations of Maxwell [24], Lewis et al. [25] and Pal [26], which assume that the filler particle can be considered spherical. The relative permeability predicted by the Maxwell equation has been added to Figure 7 for comparison. When the cuboid becomes a cube, the Maxwell correlation predicts very well the relative permeability, but fails when the cuboid is very thin and resembles a thin plate. Indeed, when thin impermeable Montmorillonite platelets are used as fillers in the polymer, the Maxwell correlation cannot be used. The Montmorillonite platelets act as a barrier material, and their associated MMMs have been considered in food packaging [12]. Wu et al. [18,19] have recently proposed a general correlation for predicting the relative permeability of MMMs embedding impermeable fillers of various shapes based on geometrical parameters of the filler rather than purely on the void fraction. At the same time, the simplicity and accuracy of the MC simulation can certainly contribute advantageously to estimating the relative permeability of MMMs for various filler geometries, as is the case for the results in Figure 7.

A series of simulations were performed with cubic filler particles (*x_F_* = *y_F_* = *z_F_*) of different sizes to change the filler volume fraction and for different solubility coefficients of the filler. At the same time, all other parameters remained the same. Figure 8a presents the results for the filler having a lower permeability than the one of the polymer matrix, such that the filler acts partially as a barrier material. Figure 8b presents the results for the filler having a higher permeability than the one of the polymer matrix such that the filler contributes to enhancing the effective permeability of the MMM by providing a preferential path for permeation. The results obtained with the MC simulations are compared to those obtained by the FD method. These results clearly show that the MC method can accurately predict the relative permeability of MMMs over a wide range of the solubility coefficient. Indeed, the average AE for the prediction of the relative permeability for the results of Figure 8a are 0.002, 0.007, 0.011 and 0.009, respectively, for the filler having a relative solubility of 0.01, 0.10, 0.20 and 0.50. Relative solubility is defined as the ratio of the filler solubility over the solubility of the polymer. The average AE for the prediction of the relative permeability for the results of Figure 8b are 0.017, 0.036, 0.040 and 0.077, respectively for the filler having a relative solubility of 2, 5, 10 and 20.

It is easier to perform MC simulations when the solubility coefficient of the filler is lower or not much higher than the solubility of the polymer matrix because the steady-state population inside the elementary unit remains at a reasonable level. When using a three-dimensional lattice grid of 41 × 41 × 41, it is recommended to have a constant upstream surface population for each cubic mesh of 40 and more to achieve a representative steady-state concentration profile. However, when the filler has a high solubility, a larger number of molecules will accumulate within the filler due to the attainment of equilibrium. As a result, the total population of molecules within the elementary unit will increase and may exceed the memory allocation currently available on many computers. When the solubility of the filler is high, the size of the filler particle that may be considered is restricted. Figure 9 presents the plot of the population of molecules as a function of the number of iterations for a cubic filler with a solubility twice the one of the polymer. This graph shows that for lower filler volume fractions, the population stabilizes below the upper memory limit. It is possible to reduce the lattice grid to a lower number of three-dimensional mesh volumes to accommodate higher filler solubility. However, this restriction is temporary as the size of memory available on most computers constantly increases.

MC simulations also allow access to the concentration profile within the MMM. Figure 10 presents the instantaneous steady-state concentration profile for a neat polymeric membrane and an MMM with an impermeable cubic filler (19 × 19 × 19) embedded in a lattice of an elementary element (41 × 41 × 41). This concentration profile is the cross-section of the median *z*-plane of the MMM. Figure 10 shows a clear concentration profile and concentration gradient despite the random nature of the permeation process. It is important to point out that this steady-state concentration profile is captured at one particular instant and the number of molecules in each mesh volume will vary from one instant to another. To attain a very well-defined concentration profile, one would need to use a much larger molecule population, provided the memory allocation and the time of computation are not excessive. Another approach is to take a moving average of the number of molecules over a period of time for each mesh volume of the elementary unit when the steady state has been achieved.

The results presented in Section 3 were obtained assuming ideal MMMs. However, for most fabricated MMMs, it is very difficult to achieve a perfectly ideal membrane because of various factors impeding this objective. The most cited reasons for not achieving ideality are particle agglomeration, polymer–filler interfacial void, polymer-chain rigidification, pore blockage and method of fabrication. It is very common in designing an MMM to assume ideality as it is impossible to forecast the defects that the fabricated membrane may suffer. The estimated performance of an MMM, obtained via popular correlations [24,25,26] for simple filler geometry or numerically by MC or FD for more complex filler geometry, is critical because it serves as a baseline in order to investigate the reasons for deviating from ideality. There are many papers where actual MMM performance was opposite to what was expected and allowed researchers postulating as to the cause of the observed discrepancies [27]. For example, Ahn et al. [28] investigated the relative permeability of MMMs prepared from polysulfone embedding various amounts of spherical nonporous fumed silica nanoparticles. They measured the relative permeability of six different gases as a function of the filler void fraction and compared their results with the Maxwell correlation [24]. Since these nonporous fumed silica nanoparticles are impermeable, they act as a barrier material and the relative effective permeability should decrease as a function of the filler volume fraction. However, this was not the case and the relative permeability increased with the filler volume fraction. They explained this discrepancy by a substantial change in the free volume in the polymer phase caused by the introduction of the silica nanoparticles. In fact, they claimed that the diffusion coefficient of the polymeric phase increased with the amount of silica embedded in the polymeric phase due to the increase in the free volume. The Maxwell correlation was used as a baseline to measure the degree of discrepancy in the relative permeability. We believe that another reason to explain this discrepancy could be due to a polymer–filler interfacial void that would enhance the diffusion coefficient around the silica nanoparticles leading to an enhancement of the permeability. The next phase of our research will use MC simulations with three phases to explain some discrepancies that are commonly observed in the literature such as interfacial void and polymer-chain rigidification. The effect of agglomeration could also be easily studied. This is where a numerical tool such as the one presented in this work becomes essential. It is simply not possible to rely on currently existing correlations.

## 4. Conclusions

This investigation considers the Monte Carlo simulation as a tool for estimating the relative permeability of ideal mixed-matrix membranes over a wide range of polymer and filler parameters. Monte Carlo simulation offers a flexible and accurate approach to estimating the relative permeability of MMMs up to high filler volume fractions. The results are comparable to those obtained by solving Fick’s second law of diffusion by finite differences. The ease of coding the MC algorithm compared to finite differences or finite elements is an enviable advantage. It was found that the MC algorithm is relatively insensitive to the scale of the grid discretization, provided that the number of molecules considered is sufficiently high. Both MC methods, MC1 and MC2, gave nearly identical results. The current MC uses the relative diffusivity for adjusting the migration of a given molecule in the different phases of the MMM, and the relative solubility to reach the equilibrium at the polymer–solid interface.

The power and memory of the current computers allow tackling important engineering problems. Some of the memory limitations encountered in this investigation will surely be resolved in the future, allowing us to consider more complex problems. We are currently extending the MC simulations to consider non-ideal MMMs in an attempt to gain a better understanding of the discrepancies observed in the literature.

## Figures and Tables

**Figure 1 membranes-12-01053-f001:**
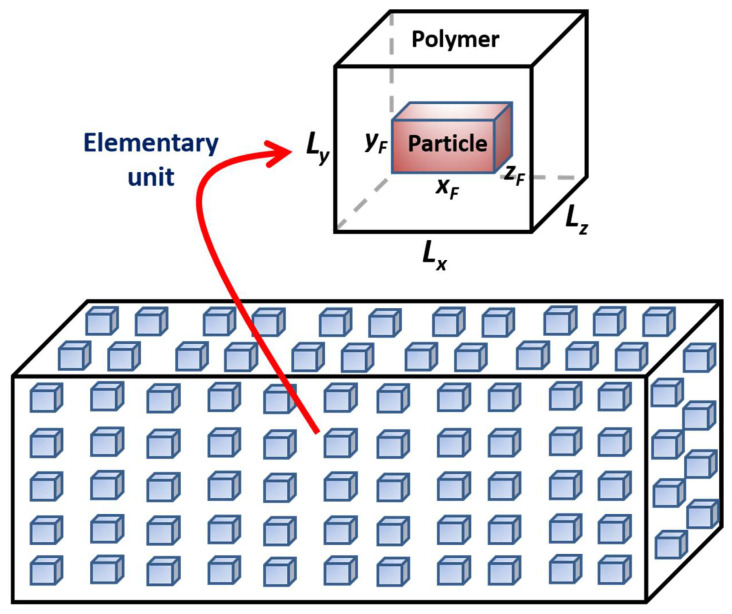
Schematic diagram of a section of an MMM consisting of a myriad of identical elementary units, each comprised of a centrally located filler particle embedded in the polymer matrix.

**Figure 2 membranes-12-01053-f002:**
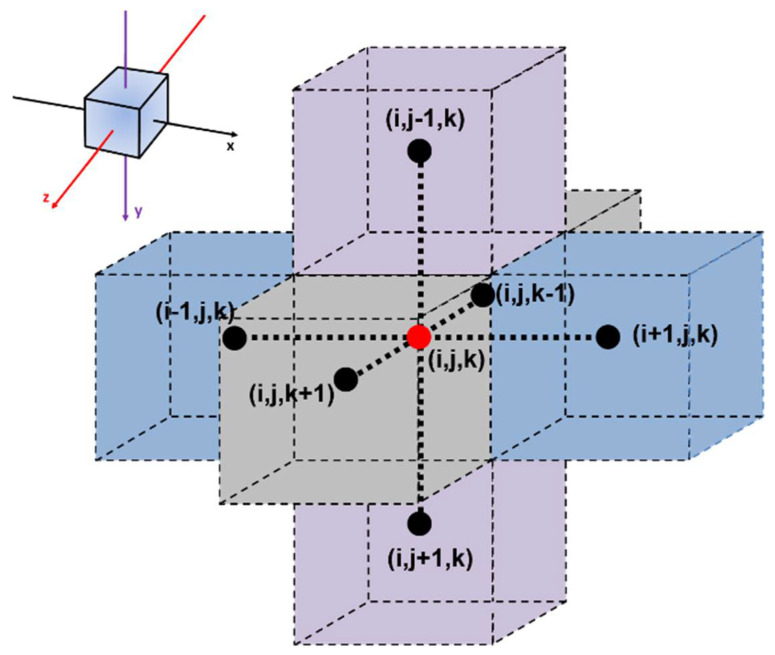
The central mesh element (*i*, *j*, *k*) surrounded by its six neighboring mesh elements as used in cubic lattice for the MC and the FD methods (Adapted from [18,19]).

**Figure 3 membranes-12-01053-f003:**
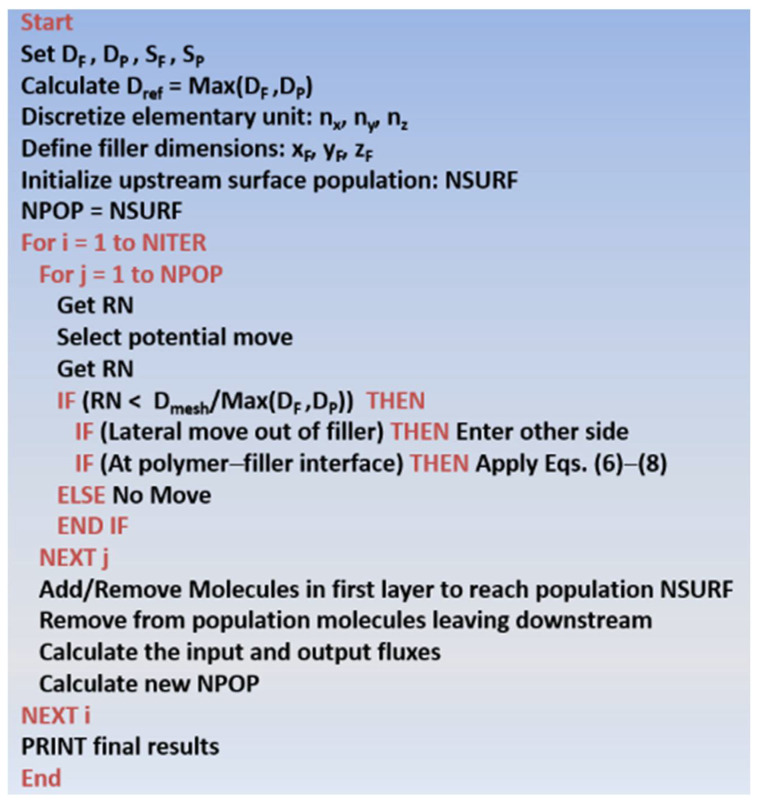
Computer pseudocode of the MC algorithm proposed in this investigation.

**Figure 4 membranes-12-01053-f004:**
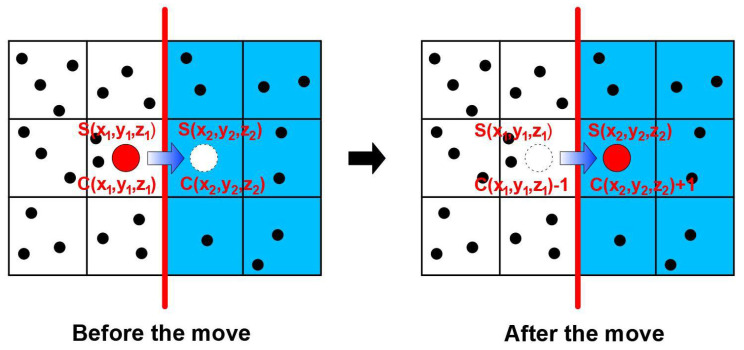
Schematic diagram of the potential migration of one molecule across the polymer–solid interface from phase 1 to phase 2 in an MMM.

**Figure 5 membranes-12-01053-f005:**
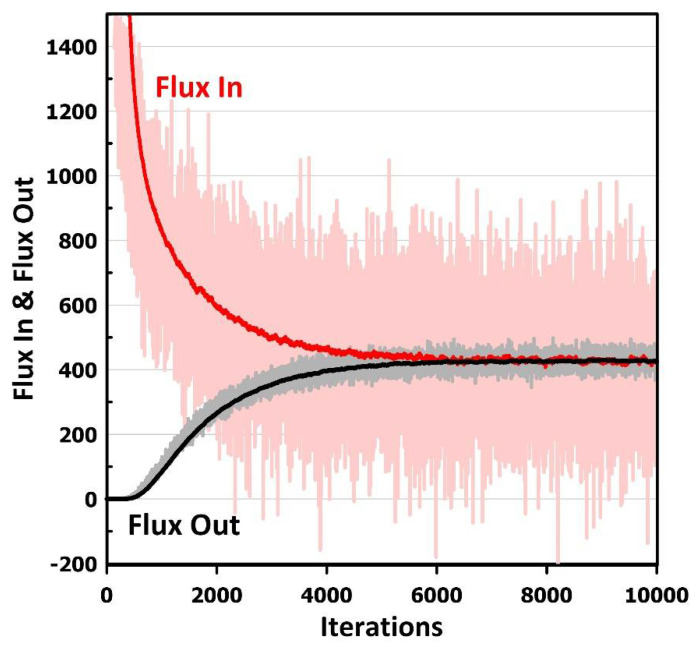
The plot of the typical fluxes in molecules at the upstream and downstream sides of the MMM using the lattice MC (MC1). The light color plots represent the instantaneous fluxes, whereas the solid lines represent the moving average fluxes. Lattice: 41 × 41 × 41; (*x_F_*, *y_F_*, *z_F_*) = (27, 27, 27); *D_F_* = *D_P_* = 1 × 10^−11^ m^2^/s; *S_F_* = 0.10, *S_P_* = 0.50.

**Figure 6 membranes-12-01053-f006:**
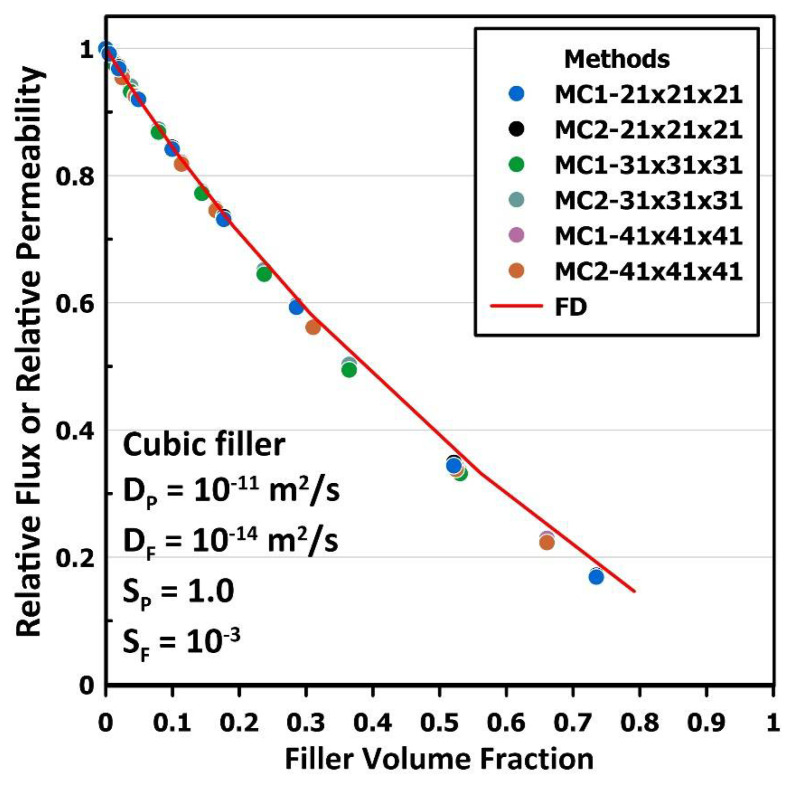
The plot of the relative steady-state flux or relative permeability of an MMM with a cubic filler particle of different sizes for the two MC methods: the lattice MC (MC1) and the continuous displacement MC (MC2). Three different discretization meshes were used for comparison, and the results were compared with the results of the FD method under identical conditions.

**Figure 7 membranes-12-01053-f007:**
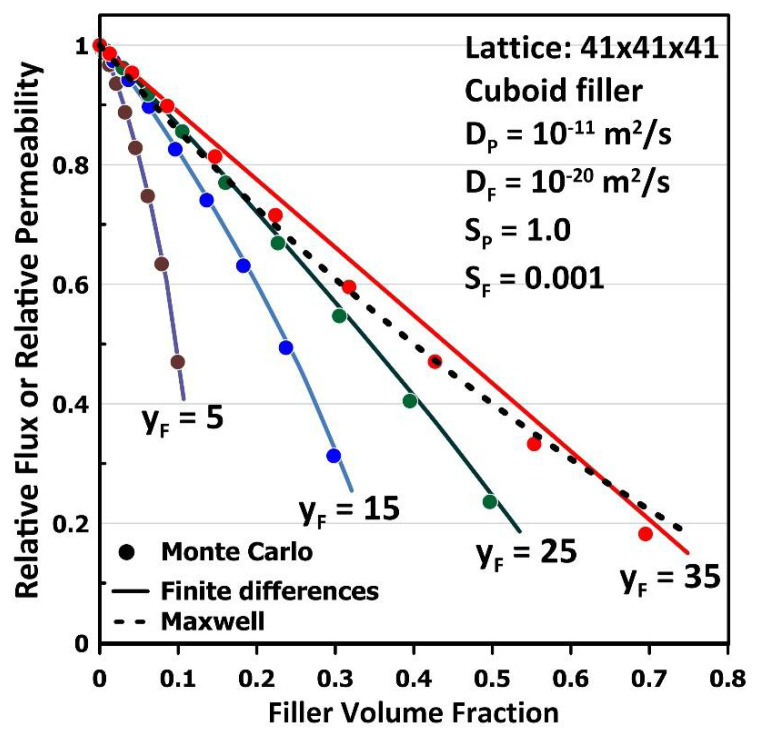
The plot of the relative steady-state flux or relative permeability of an MMM with an impermeable cuboid filler for four different thicknesses (*y_F_*) on lattice discretization of 41 × 41 × 41. The other two dimensions (*x_F_* = *z_F_*) were increased to augment the filler volume fraction. Results were compared with the results of the FD method under identical conditions.

**Figure 8 membranes-12-01053-f008:**
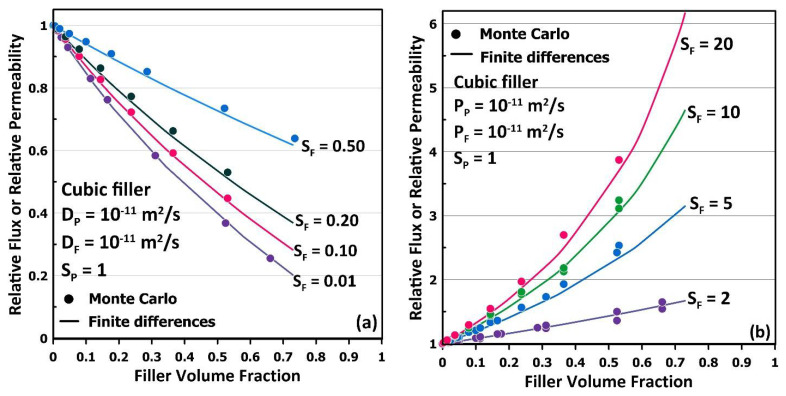
The plot of the relative steady-state flux or relative permeability of an MMM with a cubic filler particle for (**a**) four different filler solubility coefficients lower than the solubility of the polymer and (**b**) four different filler solubility coefficients higher than the solubility of the polymer. Results are compared with the results of the FD method under identical conditions.

**Figure 9 membranes-12-01053-f009:**
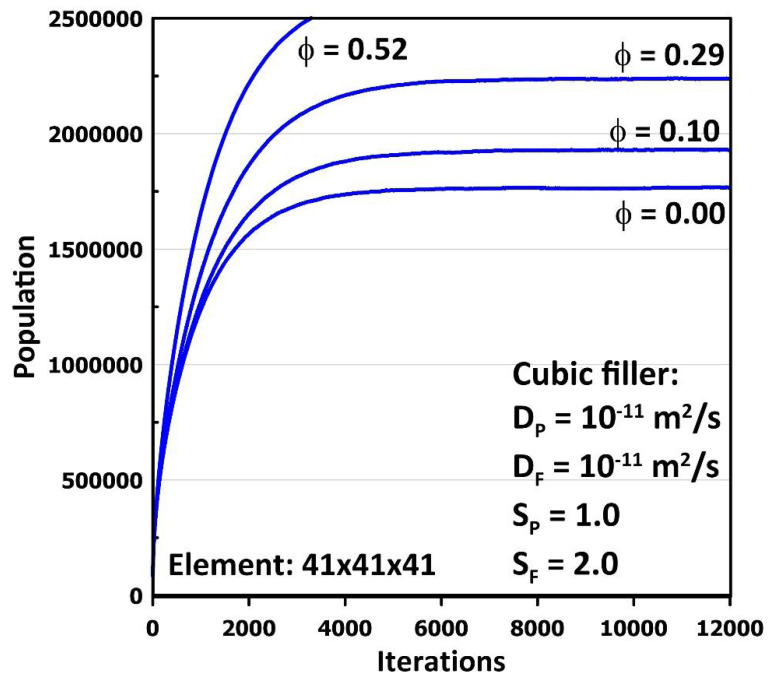
The plot of the population of molecules as a function of the number of iterations for different sizes of a cubic filler for the case where the solubility coefficient of the filler is twice that of the polymer.

**Figure 10 membranes-12-01053-f010:**
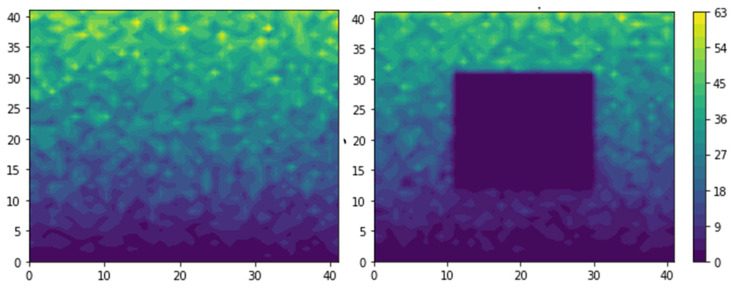
The plot of the instantaneous steady-state concentration profile for a neat polymeric membrane (**left**) and an MMM (**right**) with an impermeable filler (19 × 19 × 19) in an elementary unit (41 × 41 × 41). The *x*-*y* profile was obtained at the median *z*-direction.

## Data Availability

Data is contained within the article.

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
