# Peer review of "Monte Carlo Simulations for the Estimation of the Effective Permeability of Mixed-Matrix Membranes"

_membranes, 2022, doi:10.3390/membranes12111053_

Round 1
Reviewer 1 Report
Monte Carlo simulations for the estimation of the effective permeability of mixed-matrix membranes
Membranes
Dear editor,
The manuscript illustrated an interesting simulation with no validation with experimental results. I think the approach does need more improvements to be consistent of a simple real condition. It might have been better that the authors have submitted this manuscript after some experience in Monte Carlo simulations for their own experimental works.
Title
· The manuscript title has been written more similar to a review paper or a book. It should be rewritten to a more specialized one.
Abstract
· The abstract has been also prepared too overally.
· I couldn’t find the main characteristic(s) of the proposed MC algorithm in the abstract.
· The accuracy mentioned in abstract should be quantified.
· It could be better to describe a bit about ideal/non-ideal MMMs.
Figures
· Figure 3 should be moved to supplementary materials.
· The units of each axes of the graphs should be inserted to the figures.
Author Response
See attached report

Reviewer 2 Report
The authors propose a new modeling method based on Monte Carlo algorithm to predict gas permeabilities of mixed matrix membranes. This method is based on discretizing the MMM volume into volume elements and assigning diffusion and solubility coefficients for each volume elements. In this respect, it ignores non-idealities at the polymer/filler interface which might alter significantly gas selectivities of the MMMs, hence an analogy can be established with this approach and the Maxwell model. The authors performed a detailed parametric study to compare both the lattice MC and the continuous displacement MC to finite difference estimation, but failed to compare the accuracy of their estimation for any real MMM systems.
Hence, while the work is of potential interest to membrane community, I think it is incomplete and cannot be published without a major revision where the authors will compare the accuracy of the estimation for a number of real MMMs. I suggest that they select both ideal and non-ideal polymer/filler pairs and assess the accuracy of their method.
On the computational side, it would be useful to which extent the power and memory of the current computers are limiting factors. Are they referring to stand-alone desktop computers or clusters? What computational power did they used in these calculations?
Author Response
See attached report

Reviewer 3 Report
In this manuscript, the authors propose a robust Monte Carlo simulation as a prediction tool for the estimation of the relative permeability of ideal mixed matrix membranes (MMMs) over a wide range of polymer and filler parameters. The work is example of an intriguing area of research to increase feasibility of gas separation using advances in computational simulation.
However, I believe that the paper requires a revision (detailed below), before being considered for acceptance in the journal.
1) Abstract: Please highlight novelty of the study and provide important quantitative findings from the work.
2) Introduction: Previous related publications on state of art research for gas permeation model of MMMs should be elaborated in the Introduction section. This helps readers to grasp current progress of the research to understand content of the paper better.
3) Please define variables used in all equations clearly. If possible, it is suggested to add an abbreviation section to list out all the variables with their corresponding units.
4) Figure 2 is similar to recent published works by the authors (Wu, Kruczek, & Thibault, 2022; Wu, Thibault, & Kruczek, 2021). Please provide citation accordingly. Same goes to in-text description and equations. Please summarize and provide references if it has been adapted from author’s earlier work.
5) Please provide an overall workflow for the methodology section to summarize the algorithm in a simple and clear manner. Currently, it is not easily understood.
6) Please provide the assumptions used in developing the simulation method clearly.
7) Eq. (9) to describe average relative error seems to be not consistent with the in-text discussion in line 328.
8) Is it possible to report and compare computational time of the various methods since it is one of the major advantages of the Monte Carlo approach as highlighted by the authors?
9) Figures 6 to 8: Please provide average relative error in the figures clearly for the different methods to ease comparison.
10) The software, origin, version of the computational tool used in this work should be provided clearly.
11) Any particular validations/applications of the predictive model in estimating permeability of mixed matrix membrane from literature to demonstrate its accuracy? If yes, it is suggested to add one section for this benchmarking exercise.
References
Wu, H., Kruczek, B., & Thibault, J. (2022). A generalized model for the prediction of the permeability of mixed-matrix membranes using impermeable fillers of diverse geometry. Journal of Membrane Science, 641, 119951. doi:https://doi.org/10.1016/j.memsci.2021.119951
Wu, H., Thibault, J., & Kruczek, B. (2021). The validity of the time-lag method for the characterization of mixed-matrix membranes. Journal of Membrane Science, 618, 118715. doi:https://doi.org/10.1016/j.memsci.2020.118715
Author Response
See attached report

Round 2
Reviewer 2 Report
The authors addressed most of the comments raised by the reviewers. While I am still sceptical regading the applicability of this apporach to predict permeabilities of real MMM systems, the proposed methodology may provide a starting point for future works. Hence, its publication in this revised form would be of the interest of the membrane community.
Reviewer 3 Report
The authors have sufficiently addressed majority of the comments. The manuscript is ready to be accepted in its present form.
